# BBK32 attenuates antibody-dependent complement-mediated killing of infectious *Borreliella burgdorferi* isolates

Alexandra D. Powell-Pierce[1], Charles E. Booth, Jr.[2], Payton G. Smith[1], Brittany L. Shapiro[1], Shannon S. Allen[1], Brandon L. Garcia[2,3]*, Jon T. Skare[1]*

**1** Department of Microbial Pathogenesis and Immunology, College of Medicine, Texas A&M University, Bryan/College Station, Texas, United States of America, **2** Department of Microbiology and Immunology, Brody School of Medicine, East Carolina University, Greenville, North Carolina, United States of America, **3** Department of Biochemistry and Molecular Biophysics, Kansas State University, Manhattan, Kansas, United States of America

* jskare@tamu.edu (JTS); garciab@ksu.edu (BLG)

## Abstract

*Borreliella burgdorferi*, the causative agent of Lyme disease, has evolved unique complement evasion proteins that promote its ability to establish and maintain infection in mammalian hosts. Among these is *B. burgdorferi* BBK32, a multifunctional surface lipoprotein that binds extracellular matrix (ECM) components, including fibronectin (Fn). In addition to its ECM-binding functions, BBK32 binds to C1r, the initiator protease of the classical pathway of complement, and protects *B. burgdorferi* from complement-mediated killing following exposure to normal human serum. The disparate functions of BBK32 in adhesion and complement evasion have previously been studied in isolation. Herein we demonstrate that full-length BBK32 binds both Fn and C1 concurrently, indicating that binding of these macromolecules do not sterically hinder their simultaneous interaction. Given the link of antibody dependence to the classical pathway, we tested how the presence of BBK32 would protect infectious *B. burgdorferi* from borrelial-specific antibodies in a complement-dependent manner. BBK32 provided protection against complement activation in the presence of borrelial-specific antibodies *in vitro*. We also demonstrated, using both flow cytometry and fluorescence microscopy, that BBK32 results in the reduction of C4 deposition on the surface of borrelial cells. This work demonstrates that BBK32 can simultaneously bind to both C1r and Fn and contributes to the broader understanding of the ability of *B. burgdorferi* to evade antibody-dependent complement-mediated killing. These observations are significant as they suggest that BBK32 plays a dual role in adhesion and dissemination in infectious *B. burgdorferi*, as well as immune evasion activities, which ostensibly promotes its pathogenic potential.

**Data availability statement:** The authors confirm that all data underlying the findings are fully available without restriction. All data are presented in the manuscript and the Supporting information spreadsheet.

**Funding:** This work was supported by the Public Health Service grants AI133367 and AI146930 (to B.L.G and J.T.S) from the National Institute of Allergy and Infectious Diseases. The funders had no role in study design, data collection and analysis, decision to publish, or preparation of the manuscript.

**Competing interests:** The authors have declared that no competing interests exist.

## Author summary

Lyme disease, caused by *Borreliella burgdorferi* and other related species, is the most common arthropod-borne infection in the United States. As an extracellular pathogen, *B. burgdorferi* is exposed to the complement system—a soluble proteolytic cascade that clears microbial invaders. Complement is defined by three pathways known as the alternative, lectin, and classical. The classical pathway of complement is activated by the binding of antibodies to antigens on foreign or damaged cells. For *B. burgdorferi*, the BBK32 surface protein is known to mute the classical pathway by binding and inhibiting the initiating protease C1r. However, no studies have addressed how BBK32 protects infectious *B. burgdorferi* from borrelial-specific antibody-mediated complement clearance. Here we show that native BBK32 protects infectious *B. burgdorferi* from antibody-dependent, complement-mediated killing. Given BBK32s multifunctionality—namely its known adherence to fibronectin—we were interested to test if BBK32 could bind the C1 complex, which contains C1r, together with fibronectin. Surprisingly, our results demonstrate that these large macromolecular host molecules can bind BBK32 simultaneously. These observations suggest that the dual activity of BBK32, namely fibronectin binding and C1r inhibition, are not mutually exclusive and contribute to *B. burgdorferi*'s ability to establish infection and evade antibody-based host clearance, respectively.

## Introduction

*Borreliella burgdorferi*, the causative agent of Lyme disease, is responsible for the largest incidence of vector-transmitted illness in the United States [1–4]. Transmitted by hard ticks of the *Ixodes* genus and predominantly dependent on small rodents as a reservoir, *B. burgdorferi* can adapt to these disparate environments through a diverse set of genes regulated by environmental cues that have not yet been fully determined [5,6]. Differentially regulated genes contribute to its ability to establish infection in the vastly diverse sites it colonizes between ticks and vertebrates [6–11], as well as escape the immune systems of these hosts at distinct points in its life cycle [12,13]. During vertebrate infection, *B. burgdorferi* evades an onslaught of host immune responses, both natural and targeted, that seek to kill the spirochete [14–16]. Given that *B. burgdorferi* is largely considered an extracellular pathogen, one branch of immunity that contributes to the killing of *B. burgdorferi* during vertebrate infection is the complement cascade [17,18]. The complement system is a series of proteolytic reactions that function to recruit immune cells to the site of activation, tag the target cell with opsonins for phagocyte recognition, and create a lytic pore in the membrane of the target cell, termed the membrane attack complex (MAC) [19–22]. The classical pathway of complement is initiated by antigen-bound antibody and the resulting deposition of the C1 complex comprised of C1q, C1r, and C1s [20,21,23]. Binding of C1 to antigen-bound IgG or IgM antibodies leads to the autoactivation of the C1r initiating protease, which then cleaves C1s. Activated C1s then modifies C2

and C4 downstream via limited proteolysis that results in the deposition of C4b and C2a to form a functional C3 convertase that then tags the target cell with C3b. The linkage of C3b to the target cell results in opsonization or the formation of the membrane attack complex (MAC) [20,21]. During tick transmission of *B. burgdorferi* in mice, the presence of murine natural IgM—that are not directed toward any particular invader, including *B. burgdorferi*—minimizes the number of spirochetes in the tick midgut, presumably due to complement-mediated killing [24]. Nonetheless, serum from mice previously infected with *B. burgdorferi* is more effective at spirochete destruction than naïve serum, implying a role for both natural and directed antibody responses in reducing *B. burgdorferi* numbers at different stages during transmission and infection [24,25]. The ability of *B. burgdorferi* to quell the classical complement cascade, following antibody recognition, would seemingly result in extended survival and persistence in the face of both a non-specific and specific humoral response.

*B. burgdorferi* encodes for a wide range of complement inhibitors that function to prevent killing mediated by the complement cascade, with significant work focused on the alternative pathway and, more recently, the classical pathway [26–28]. Resistance to the alternative pathway *by B. burgdorferi* is mediated by numerous surface-exposed proteins designated as CRASP, OspE, Erp, or Csp proteins [29–35]. These proteins facilitate the binding of factor H to the surface of the borrelial cells thereby preventing the activation of C3 convertase and are produced within *B. burgdorferi* at different stages of the zoonotic cycle [36,37]. For the classical pathway, several borrelial proteins have been identified that inhibit activation. For example, p43, a protein of unknown identity that functions as a recruiter of C4 binding protein, inhibits both the classical and lectin pathways [26–28,38]. Separately, OspC recognizes C4b, inhibiting C3 convertase formation [39]. Although p43 or OspC do not prevent C4 cleavage, they do impair deposition of C3b, a crucial component of complement, preventing the activation of the terminal components of complement. More recently, proteins from the OspE/F-like leader peptide family (Elps) have been shown to block the classical pathway by preventing C1s-mediated cleavage of C4 and C2 [40–42]. BBK32, which is upregulated during vertebrate infection, can potently block activation of the classical pathway by binding to C1r [36,37,43–46].

Prior to its discovery as a complement inhibitory protein, BBK32 was characterized as a fibronectin (Fn) binding protein, and then for its ability to bind glycosaminoglycans (GAGs) [47–51]. These ECM interactions are thought to play a role in *B. burgdorferi*'s ability to extravasate through the vascular endothelium and disseminate throughout the infected host, mediating its ability to infect and colonize distal tissues [47,51,52]. During murine infection studies, a *bbk32* mutant exhibits attenuated infectivity at lower infection doses [47,52]; however, it is not clear if this phenotype is associated with the loss of BBK32 binding to ECM (GAGs or Fn) or due to abrogated BBK32-associated complement inhibitory activity. Additionally, the role of targeted antibodies in *B. burgdorferi* classical pathway-mediated killing, broadly and in the scope of BBK32's activity, has not yet been established. Because the classical pathway is dependent on antibody binding to the *B. burgdorferi* surface for its activity, BBK32-mediated inhibition may be important in both the initial stages of pathogen clearance when "natural" IgM recognize *B. burgdorferi* and later once a targeted antibody response has developed [24,25]. Antibody binding can lead to a wide range of immune-mediated pathogen clearance mechanisms, from phagocyte binding and phagocytosis to activation of lymphocytes [20,21]. Previously, we have shown that key residues of BBK32 occlude the C1r active site within its catalytic serine protease (SP) domain [44]. When specific residues are mutated from their native amino acids to alanine, BBK32's binding to C1r and classical complement inhibitory activity is lost [44].

Earlier studies focused on identifying key regions of BBK32 interfacing with Fn and GAG, which are likely important for adhesion to vasculature during the dissemination phase of Lyme disease [53,54]. Both the GAG and Fn binding domains map to the intrinsically disordered amino terminal half of BBK32 [46,48–51,55]. The known interactions between BBK32 and its ligands Fn, GAG, and C1r have been studied independently *in vitro* [46,48–51,55]. A prior study showed that BBK32's Fn- and GAG-binding activity served to slow the spirochetes in the vasculature, thus promoting extravasation [53,54]. Large domain deletions in BBK32, when analyzed in mouse vasculature using intravital microscopy, showed that BBK32 functions to first tether *B. burgdorferi* to the vasculature via its Fn interactions, functioning as a brake, while GAG-binding positions the cells optimally for extravasation through longer-lived interactions [54]. Despite this, the

spatiotemporal role of each function during infection, in particular BBK32's C1r-binding activity, remains unclear. Additionally, the ability of BBK32 to interact with ECM components and C1r or the entire C1 complex, either independently or simultaneously, has not been determined.

Here, we use protein structural modeling and biophysical analyses to show that BBK32 can simultaneously interact with Fn and C1. Using an *in vitro* model to demonstrate the antibody-dependence of complement killing in an infectious isolate of *B. burgdorferi*, we also demonstrate the ability of BBK32 to partially protect borrelial cells from antibodies directed against *B. burgdorferi*. As a result of these studies, we have determined that BBK32's adhesin and C1r inhibitory activity may play a role not only during initial stages of infection, but its ability to impair classical complement-mediated clearance after a robust borrelial antibody response has been mounted. In addition, these studies have highlighted BBK32's ability to interface with both Fn and C1, suggesting that these seemingly disparate roles occur concurrently to facilitate dissemination and survival within an infected host.

## Materials and methods

### AlphaFold3 analysis

The AlphaFold3 (AF3) server [56] was used to predict a ternary complex of BBK32, Fn and C1r. Amino acid sequences corresponding to the Fn domains $^{2-5}$FnI (UNIPROT: P02751, residues 95–273), the C1r SP domain (UNIPROT: P00736, residues 464–702) were co-folded with BBK32 (UNIPROT: O50835, residues 131–354). Structural alignments and root mean square deviation (rmsd) calculations were performed with Pymol (The PyMOL Molecular Graphics System, Version 3.0 Schrödinger, LLC) using the previously resolved crystal structures of the unbound C-terminal region of BBK32 (PDB: 6N1L, [45]), BBK32 in complex with C1r CCP1-CCP2-SP (PDB: 7MZT, [44]), and BBK32 in complex with $^{2-3}$FnI (PDB: 4PZ5, [57]).

### Bacterial strains and plasmids

*Borreliella burgdorferi* B31 strains ML23/pBBE22*luc*, B31-A3-GFP [58,59], and the non-infectious derivative B314, were grown in BSK-II and 6% normal rabbit serum (Pel-Freez Biologicals, Rogers, AR) as described [60,61]. The *bbk32* mutant derivatives of ML23 and B31-A3-GFP, JS315/pBBE22*luc* and GP100, respectively, are referred to in Table 1. B31-A3-GFP and GP100 were cultured with gentamicin at 50 μg/ml under conditions that partially mimic the mammalian environment (5% $CO_2$, 37°C, pH 6.8). ML23/pBBE22*luc* and JS315/pBBE22*luc* were grown under similar mammalian-like conditions in kanamycin at 300 μg/ml. Additional antibiotic resistance for JS315 and GP100 are indicated in Table 1. All *B. burgdorferi* strains containing either pBBE22*luc*, pCD100, or pAP7 were grown with kanamycin at 300 μg/ml.

### Transformation of *B. burgdorferi*

Transformation of strains *B. burgdorferi* ML23 with the plasmid construct pAP7, and B31-A3-GFP with pCD100 and pAP7 was performed as previously described [44,47,63–65]. The presence of plasmids pCD100 and pAP7 was selected in complete BSK-II media using kanamycin at a final concentration of 300 μg/ml. As is the norm, all borrelial transformants were screened for their composition of all plasmid DNA [58]. Only strain transformants that maintained the collection of plasmids found in their parental derivative were used.

### Far Western overlay analysis

Far Western overlays were carried out essentially as described [46,63], using *B. burgdorferi* strains ML23/pBBE22*luc* and B31-A3-GFP along with their respective *bbk32* mutant derivatives (JS315/pBBE22*luc* and GP100, respectively; see Table 1). The *bbk32* mutant derivatives (JS315 alone and GP100) were transformed with either pCD100 or pAP7, which encode for native *bbk32* or the *bbk32*-R248A/K327A allele, respectively, expressed using the native *bbk32* promoter (Table 1). Whole-cell lysates were generated for the borrelial strains, and $2.5 \times 10^7$ whole cell equivalents were resolved by SDS-PAGE and then

**Table 1.** *Borreliella burgdorferi* strains used for present study.

| *Borreliella burgdorferi* Strain | Description | Reference |
|---|---|---|
| ML23/pBBE22*luc* | Serum-sensitive, non-infectious *B. burgdorferi* B31 derivative strain lacking linear plasmid 25 with shuttle vector pBBE22 encoding *bbe22* and *B. burgdorferi* codon-optimized *luc* gene under the control of a strong borrelial promoter (P$_{flaB}$-*luc*); kan$^R$. | [52,62] |
| JS315/pBBE22*luc* | *bbk32* mutant in ML23-pBBE22*luc* background; strep$^R$, kan$^R$. | [52] |
| JS315/pCD100 | JS315 with wildtype *bbk32* under control of the *bbk32* native promoter in pBBE22*luc*; strep$^R$, kan$^R$. | This study |
| JS315/pAP7 | JS315 with *bbk32* R248A/K327A under control of the *bbk32* native promoter in pBBE-22*luc*; strep$^R$, kan$^R$. | This study |
| B31-A3-GFP | B31-A3 transformed with *gfp* under control of the borrelial *flaB* promoter on the cp26 plasmid; gent$^R$. | [59] |
| GP100 | *bbk32* mutant in B31-A3-GFP background; gent$^R$, strep$^R$. | This study |
| GP100/pCD100 | GP100 with wildtype *bbk32* under control of the *bbk32* native promoter in pBBE22*luc*; gent$^R$, strep$^R$, kan$^R$. | This study |
| GP100/pAP7 | GP100 with *bbk32* R248A/K327A under control of the *bbk32* native promoter in pBBE-22*luc*; gent$^R$, strep$^R$, kan$^R$. | This study |
| B314/pCD100 | Non-infectious *B. burgdorferi* strain B31 derivative missing all linear plasmids transformed with intact *bbk32* (with *bbk32* promoter) cloned into pBBE22*luc*; kan$^R$ | [44,46,52] |
| B314/pAP7 | Non-infectious *B. burgdorferi* strain B31 derivative missing all linear plasmids transformed with the *bbk32*-R248A/K327A allele (under the control of the *bbk32* promoter) cloned into pBBE22*luc*; kan$^R$ | [44,52] |

transferred to PVDF membranes. Membranes were blocked overnight in 5% non-fat milk, washed, and then incubated with either (1) 20 µg human fibronectin (EMD Millipore); (2) 20 µg active human C1r (Complement Technologies). Membranes were then washed and probed for either fibronectin or C1r using an anti-human Fn HRP conjugate (Santa Cruz Biotechnology), or an anti-human C1r HRP conjugate (Santa Cruz Biotechnology), respectively (both diluted 1:5,000). To confirm BBK32 production, a monoclonal antibody to the C-terminus of BBK32 (contracted to and produced by ProMab Biotechnologies) was used at a 1:10,000 dilution, and the samples normalized to FlaB levels using an anti-borrelial FlaB monoclonal antibody (US Biological). Both monoclonal antibodies were detected using a goat anti-mouse immunoglobulin HRP conjugate diluted 1:10,000 (ThermoFisher Scientific). Blots were then visualized using the Western Lightning Plus-ECL (PerkinElmer).

## C1r and Fn binding studies

Surface plasmon resonance (SPR) assays were performed using immobilized BBK32 on a CMD200 sensor chip (Xantec) via standard amine coupling using conditions previously described [44–46,63]. Either 20 nM Fn or 50 nM C1, or a mixture of 20 nM Fn and 50 nM C1, were individually injected over the BBK32 biosensor for 2 min, followed by a dissociation time of 3 min. Two 60 s injections of regeneration buffer (0.1 M Glycine (pH 2.0), 2.5 M NaCl) were used to return the biosensor to baseline. Purified C1 was obtained from Complement Technologies. Purified human fibronectin was obtained from Millipore Sigma. BBK32 was purified according to previously published protocols [50].

The ability of BBK32 to concurrently interact with Fn and C1 was determined based on the binding response just prior to the injection stop and calculated by subtracting the Fn sensorgram from the co-injection sensorgram to calculate residual binding. The resulting residual binding sensorgram was then compared to the injection of C1 alone.

## Flow cytometry

Two borrelial strains in the non-infectious, serum-sensitive strain B314 background were evaluated by flow cytometry using either native *bbk32* (encoded by pCD100; [44,46]) or the *bbk32*-R248A-K327A allele that is unable to bind or inhibit C1r

(encoded by pAP7; [44]). The cells were grown to mid log phase, washed in PBS, and incubated with 50 µg octadecyl-rhodamine to fluorescently label the cells for 30 minutes at 37°C. The cells were centrifuged at 3600 x $g$ and washed twice to remove unincorporated octadecyl-rhodamine. Cells were then incubated for 30 minutes with 175 ng rabbit anti-*B. burgdorferi* polyclonal antibody (Abcam). C5-depleted human serum (Complement Technologies) was added as a source of complement to a final concentration of 10%. Following fixation, the samples were incubated for 30 minutes with 1 µg of a mouse mono-clonal antibody to C4c (Quidel). Cells were washed with PBS with 0.5% BSA and then incubated for 30 minutes with 1 µg anti-mouse IgG Alexa Fluor 647 to detect C4c deposition. Cells were washed with PBS prior to flow cytometry.

Each group of fixed cells were then analyzed using the BD Fortessa X-20 within 24 hours of sample preparation. 10,000 events were read for each sample on the flow cytometer. Cells were characterized based on their individual fluorescent properties and analyzed with BD FACSDiva and FlowJo Software. *B. burgdorferi* strain B314-pCD100 were also prepared following the same protocol described but without staining, were individually stained with octadecyl-rhodamine, or with secondary antibody for compensation of spectral overlap. To establish proper gating and demonstrate primary antibody specificity, events were first gated on forward scatter (FSC) and side scatter (SSC). Rhodamine-positive cells were then gated for the presence of C4c. B314-pCD100 without C4c antibody treatment was used to gate negative C4c events with a 0.2% false positive rate.

## Fluorescent microscopy and colocalization analysis

Colocalization of anti-*B. burgdorferi* antibodies and complement component C4c was determined using the *B. burgdorferi* B31-A3-GFP strain derivatives after washing the cells and incubating with 1 µg of a rabbit anti-*B. burgdorferi* polyclonal antibody or a rabbit isotype control (Abcam). C5-depleted human serum (Complement Technologies) was added to 20% as a source of complement. Subsequently, cells were incubated for 30 minutes with 0.5 µg of a mouse monoclonal antibody to C4c (Quidel). Cells were washed with PBS with 0.5% BSA and then incubated for 30 minutes with 1 µg anti-rabbit IgG-Alexa Fluor 594 Plus to detect added rabbit antibodies, as well as 1 µg anti-mouse IgG-Cy5 to detect bound C4c antibodies. Cells were then fixed using 4% methanol-free paraformaldehyde and mounted onto a slide using Prolong Diamond Antifade Mountant (ThermoFisher). Cells were imaged using the Olympus Fluoview FV3000 confocal microscope, with equivalent exposure times for each fluorophore channel across samples. Cells were also prepared without antibody treatment and imaged in the GFP and DIC channels to demonstrate that all cells analyzed were GFP positive.

Colocalization analyses to determine the intensity correlation quotient (ICQ) were performed using the JACOP intensity correlation coefficient-based analysis plugin for ImageJ [66] separately for five biological replicates per sample using both antibodies directed against *B. burgdorferi* and an isotype control.

## Antibody-dependent complement-mediated killing assays

Antibody-dependent complement killing was assessed *in vitro* by diluting cells to a concentration of 5 x 10⁶ *B. burgdorferi* cells/ml in PBS, 0.5% BSA. Cells were then exposed to 175 ng rabbit anti-*B. burgdorferi* polyclonal antibody (Abcam) in addition to 20% NHS (Complement Technologies) for 30 minutes similar to recently published work [42]. Control groups were exposed to 175 ng rabbit isotype antibody (Abcam) coupled with 20% NHS to depict the antibody-dependent binding associated with complement activation, or 175 ng anti-*B. burgdorferi* antibody coupled with heat inactivated NHS to test whether the killing was purely antibody-dependent and assess the complement-dependence on borrelial death, respectively. Cells were then assessed for viability using a dark field microscope based on motility and membrane disruptions, as done in earlier published work [44,63].

## Statistics

Data is shown as the average of the replicates with 95% confidence intervals. A one-way ANOVA with a Tukey's multiple comparisons test was used for SPR residual binding analysis. For the flow cytometry analysis, a two-tailed unpaired t-test

was used. For antibody-dependent killing assays, two-way ANOVA with a Šidák correction for multiple comparisons was used. All statistical analysis was performed using GraphPad Prism version 9.3.

## Results

### BBK32 binds both fibronectin and C1 concurrently *in vitro*

To assess the multifunctionality of BBK32, we determined if the amino-terminal Fn binding and carboxy-terminal C1r inter-actions were mutually exclusive. Previous work assessing the Fn-binding domain of BBK32 determined that amino acid residues 131–162, intrinsically disordered until bound to Fn, were necessary for binding to human fibronectin [49–51,54,57,67–70]. Residues 206–354 were deemed BBK32's "ordered" region, forming a stable four-helix bundle structure that binds tightly to C1r [44–46,50].

We hypothesized that BBK32 might be able to concurrently bind to both Fn and C1r, allowing for both its extracel-lular matrix binding and complement inhibitory activities during *B. burgdorferi* infection. Using AlphaFold3 (AF3) [56], predictions of each protein-protein interaction were made using residues from BBK32 previously shown to be important for each interaction (residues 131–354) along with the relevant domain truncations of each host protein (*i.e.*, $^{2-5}$FnI and C1r SP) (Fig 1). The resulting model was of high quality as judged by consistently high pLDDT scores across the model (*i.e.*, > 70), a high pTM score (*i.e.*, 0.73), a moderately high iPTM score (*i.e.*, 0.71), and low PAE errors for BBK32 residues and the corresponding Fn or C1r residues (S1 Fig) [56]. Furthermore, the resulting model of the ternary complex was consistent with previously resolved experimental structures in three important ways: i) the C-terminal region of the BBK32 AF3 model closely matches the crystal structure of BBK32-C (RMSD = 0.181 Å; PDB: 6N1L ( [45]); ii) the binary complex of C1r/BBK32 aligns closely with the co-crystal structure of BBK32-C in complex with a proteolytic fragment of C1r (RMSD = 0.586 Å; PDB: 7MZT ( [44]); and iii) similar agreement is found between the binary prediction of Fn/BBK32 and the crystal structure of an N-terminal BBK32 peptide in complex with domains $^{2-3}$FnI of Fn (RMSD = 0.737 Å; PDB: 4PZ5 ( [57]).

The AF3 model presented in Fig 1 predicts that it is structurally feasible for full-length BBK32 to interact with domain fragments of each host ligand. However, given the large size of each full-length host molecule (*i.e.*, Fn (~500 kDa dimer) and C1 (~760 kDa as $C1qC1r_2C1s_2$), we sought biophysical evidence that these interactions can occur *in vitro*. Here we used purified recombinant full-length BBK32 to produce surface plasmon resonance (SPR) biosensors. Purified human C1 (50 nM), purified human Fn (20 nM), or a co-injections of 20 nM Fn and 50 nM C1 were then injected over immobilized full-length BBK32. The resulting sensorgram for 20 nM Fn was then subtracted from the co-injection curve to calculate residual C1 binding by the BBK32 surface (Fig 2A and 2B). Analysis of the residual binding responses strongly suggested that BBK32 can bind both host proteins simultaneously *in vitro* (Fig 2C).

### BBK32 C-terminal double alanine mutant binds Fn but not C1r

We have previously shown that BBK32's ability to bind and inhibit human C1r is abrogated by mutations introduced within its complement inhibitory domain that map to its C-terminus [44–46]. The targets of our prior studies were amino acid residues R248 and K327, which interact with the C1r active site and B loop, respectively (also referred to as the K1 and K2 sites recently [43,44]). In our previous work, assays were performed in the ML23 background as we had used this strain in the *bbk32* mutant infectivity analyses [47,58]. However, the ML23 derivative lacks the linear plasmid 25, which is essential to complete the enzootic cycle [58,71]. In addition, screening a distinct *B. burgdorferi bbk32* mutant in an independent infectious derivative provides additional evidence that the results observed were not strain specific. For comparison we used the B31-A3-GFP and the ML23 (Fig 3) genetic backgrounds to independently assess subsequent overlays and cell sensitivity assays. To assess the ability of wild type BBK32 and the double alanine (DA) mutant (BBK32-R248A/K327A) to bind to C1r and Fn, we performed Western blots first showing that

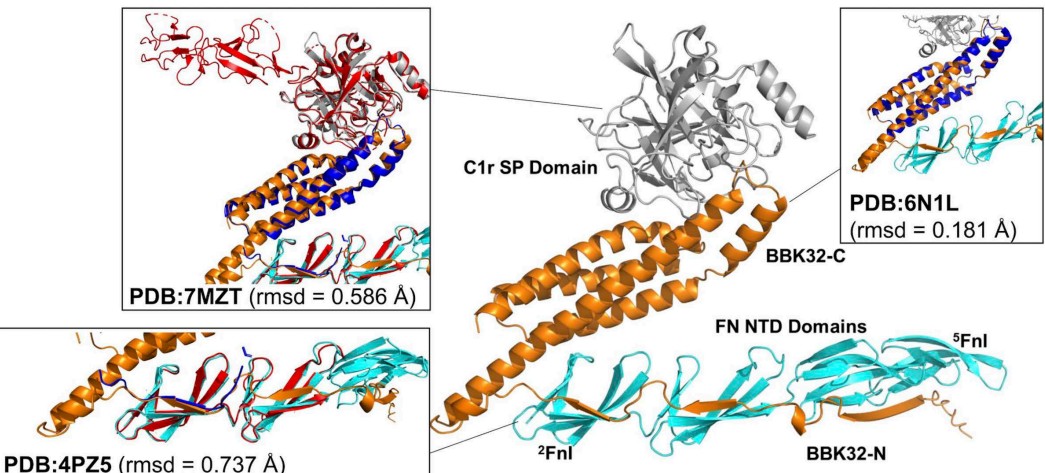

**Fig 1. AlphaFold3 model of the C1r-BBK32-Fn ternary complex.** AlphaFold3 was used to simultaneously fold BBK32 (orange, residues 131–354; UNIPROT: O50835), the C1r serine protease (SP) domain (grey, residues 464–702; UNIPROT: P00736), and the Fibronectin N-terminal domains (FN NTD; [2]FnI-[5]FnI) (cyan, residues 95–273, UNIPROT: P02751). Structural alignments to the published crystal structures of BBK32-C (PDB: 6N1L), the complex of BBK32-C and C1r (PDB: 7MZT), and the complex of BBK32-N with [2]FnI-[3]FnI (PDB: 4PZ5) are shown inset. C1r and Fn from the crystal structures are drawn in red in each alignment, while BBK32 residues are shown in blue. Root mean square deviation (rmsd) (αC) is shown for each alignment.

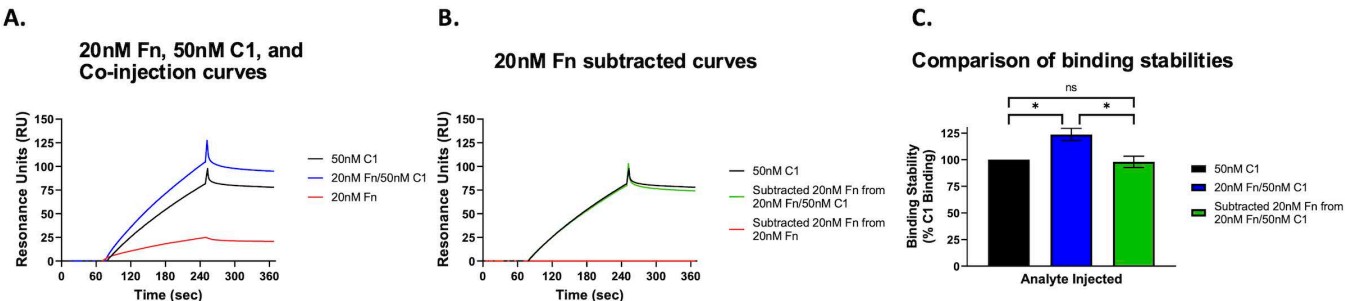

**Fig 2. BBK32 binds C1 and Fn concurrently.** A. SPR assays were performed using immobilized BBK32 injected with 20 nM human Fn, 50 nM human C1, and B. a co-injection of 20 nM human Fn and 50 nM human C1. C. The ability of BBK32 to concurrently interact with human Fn and C1 was calculated by subtracting the human Fn curve from the co-injection curve and comparing that curve to the human C1-alone injection curve. Three technical replicates were performed. Error bars represent SD. Statistical significance of the residual binding signals were assessed using a one-way ANOVA with a Tukey's multiple comparisons test, * $P < 0.002$, ns = not significant.

protein lysate collected from parent strains (ML23 pBBE22*luc* and B31-A3-GFP), the *bbk32* wild type (WT) complement (JS315 pCD100 and GP100 pCD100), and the *bbk32* DA mutant complement (JS315 pAP7 and GP100 pAP7) produced BBK32, while the *bbk32* knockout strains (JS315 pBBE22*luc* and GP100) did not. Overlays were performed probing these protein lysates with human enzyme C1r and Fn (Fig 3). In these assays, the parent, native *bbk32* complement, and the *bbk32*-R28A/K327A allele (DA mutant) complement bound Fn, while only the parent and WT complement bound C1r, indicating that the DA mutations in the C-terminus did not affect binding of Fn in the N-terminal portion of BBK32. This targeted abrogation of BBK32's C-terminal C1r binding activity allows for pointed functional analyses of BBK32's distinct activities.

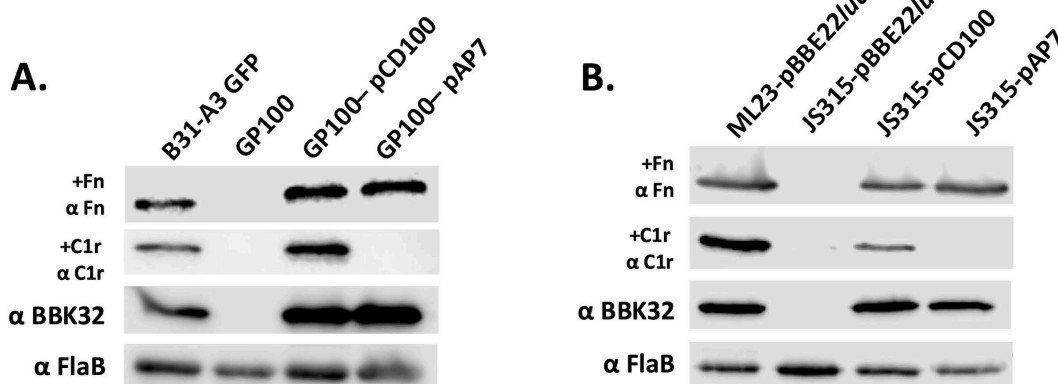

**Fig 3. Differential binding of human Fn and C1r to *B. burgdorferi* strain B31-A3, *bbk32* mutant, and functional and non-functional *bbk32* derivatives.** A. Overlay experiments performed using B31-A3 GFP parent, and isogenic *bbk32* mutant strain (GP100), native *bbk32* complement (GP100 pCD100), and a *bbk32*-R248A/K327A double alanine (DA) mutant complement (GP100 pAP7). B. Identical overlays for strain ML23 pBBE22*luc* (parent), the *bbk32* mutant derivative (JS315 pBBE22*luc*), native *bbk32* complement (JS315 pCD100), and a *bbk32*-R248A/K327A double alanine (DA) mutant complement (JS315 pAP7). For both sets of strains, protein lysates were incubated with human Fn or human C1r and then probed separately with antibody reagents specific for either human Fn or C1r (see Methods). These same samples were also probed with monoclonal antibodies to BBK32 and *B. burgdorferi* FlaB (third row and bottom row, both panels, respectively).

## BBK32 reduces complement activation on the surface of *B. burgdorferi*

The DA mutant phenotype was used to address the role of borrelial-specific antibody in the complement inhibition function of full-length BBK32 by quantifying C4c deposition on the surface of spirochetes using flow cytometry. The degree of complement activation was tested in a serum-sensitive derivative of *B. burgdorferi*, strain B314, that we have used previously to assess resistance to normal human serum [44–46]. We used a strain that made intact BBK32 (B314 pCD100; Fig 4A) and one that carries the *bbk32*-R248A-K327A allele that does not bind or inhibit human C1r (B314 pAP7; Fig 4B) [44]. Anti-*B. burgdorferi* antibody was added with C5-depleted human serum and cells were screened for C4c deposition by flow cytometry (Fig 4). C5-depleted serum was used to prevent the formation of the membrane attack complex (MAC) that results in cell damage and difficulty in gating the cells by flow cytometry. A control with no added antibody to C4c was also included as indicated (see Methods; Fig 4C). Fig 4D shows the results from cells encoding native BBK32 (B314 pCD100) relative to the DA BBK32 mutant that cannot bind or inhibit C1r (B314 pAP7). Cells producing wildtype BBK32 exhibit significantly lower levels of C4c deposition than cells that produce the BBK32-R248A/K327A mutant. These results are consistent with functional BBK32 inhibiting C1r and concomitantly reducing C4 proteolysis and subsequent attachment of C4c to the borrelial cells that encode it.

## BBK32 protects an infectious isolate of *B. burgdorferi* from borrelial-specific antibody-mediated complement-dependent killing *in vitro*

To determine whether BBK32 inhibits the deposition of classical complement components on the surface of infectious isolates of *B. burgdorferi*, we exposed borrelial cells to borrelial-specific antibodies and scored for surface deposition of C4c—as a proxy for C4b—using an immunofluorescent microscopy readout using GFP-producing cells (Fig 5). All detectable *B. burgdorferi* cells for these strains produce GFP as shown in S2 Fig. As with the flow cytometric analysis, we used C5-depleted human serum as the source of complement to reduce the damage of *B. burgdorferi* cells via formation of the MAC while maintaining a readout for classical complement activation and inhibition. Following borrelial antibody incubation and exposure to C5-depleted human serum, the parent strain (B31-A3-GFP) and *bbk32* mutant that expressed

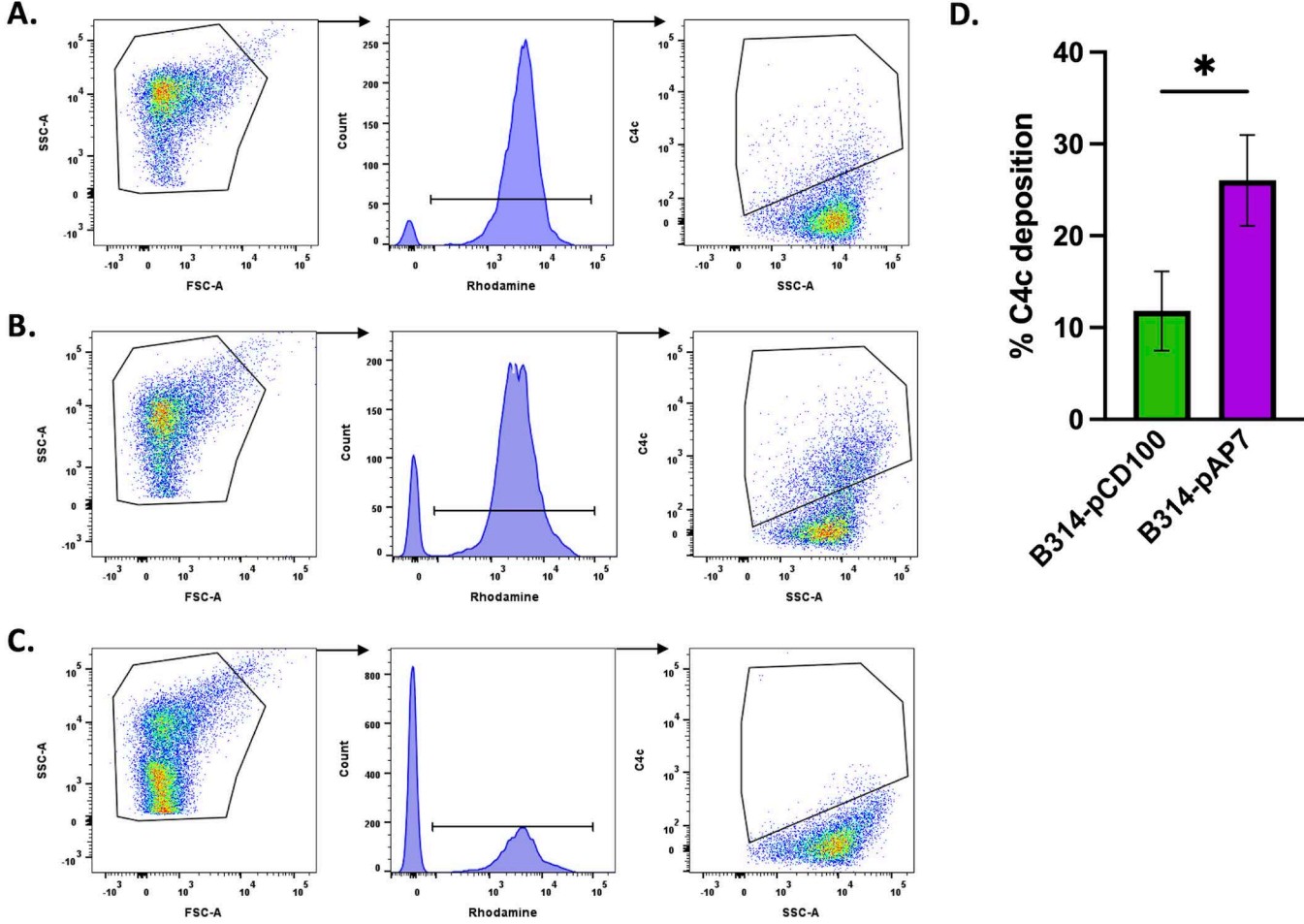

**Fig 4. Loss of BBK32 complement binding and inhibition results in increased C4 deposition.** The serum-sensitive derivative B314 was trans-formed with either native *bbk32* or the *bbk32*-R248A/K327A allele that lacks C1r binding or inhibition activity. Both cells were incubated with antibodies against *B. burgdorferi*, followed by C5-depleted human serum. The level of C4c deposition was then scored for 10,000 events using flow cytometry tracking the deposition of C4c on the surface of the spirochetes. The isolate encoding wildtype *bbk32* is shown in Panel A, the strain with *bbk32*-R248A/K327A (double alanine mutant [DA]) is depicted in Panel B, and a control with no antibody to C4c is indicated in Panel C (see Methods for detail). Panel D shows the data comparing wildtype *bbk32* (green) relative to expressing *bbk32*-R248A/K327A (purple) in *B. burgdorferi* strain B314 as a histogram plot with the background binding shown in Panel C subtracted. The presence of wildtype BBK32 reduces the amount of C4c deposition relative to the BBK32-R248/K327A DA mutant consistent with the mutant's inability to bind and inhibit C1r and reduce classical complement activation. The mean for three individual replicates is shown with standard deviation. * P = 0.02.

native *bbk32* (GP100 pCD100) had reduced C4c localized on their surface (Fig 5). In contrast, *B. burgdorferi* cells lacking BBK32 (GP100), and particularly those expressing the *bbk32*-R248A-K327A allele in a *bbk32* mutant background (GP100 pAP7), showed more C4c when incubated with the anti-*B. burgdorferi* antibody consistent with their reduced BBK32 complement inhibitory activity (Fig 5). However, when a rabbit isotype control antibody was combined with the C5-depleted serum, all cells, independent of genetic composition, showed less C4c deposition consistent with reduced control antibody binding and concomitant decreased complement activation (S3 Fig). To address the degree of C4c deposition, we scored for colocalization of GFP with C4c using integrated colocalization quotient (ICQ) calculations between these samples [66] (S4 Fig). While only the *bbk32* complement strains demonstrated a significant difference with this assessment, the overall trend was consistent with reduced classical complement resistance activation only in cells with functional BBK32. The

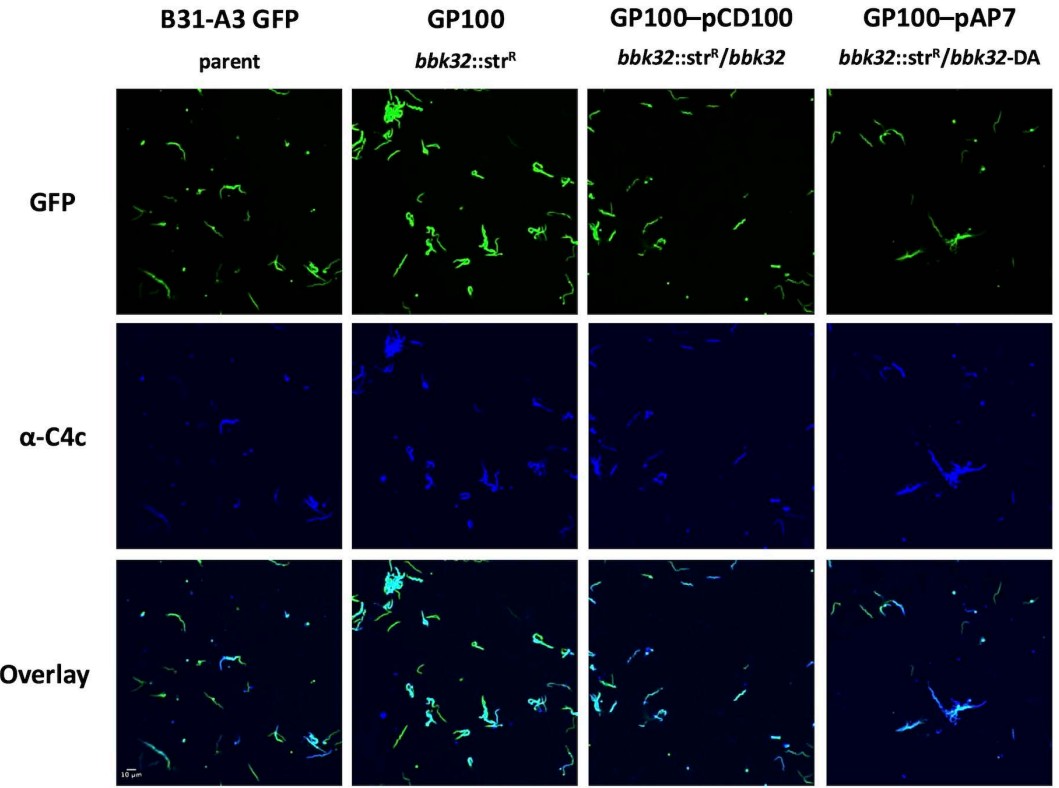

**Fig 5. Qualitative assessment of increased deposition of complement components in *bbk32* mutant strains.** Green fluorescent protein (GFP) producing *B. burgdorferi* strains B31-A3 GFP, GP100 (B31-A3 GFP *bbk32*::Str^R), GP100 pCD100 (*bbk32*::Str^R with native *bbk32* complement), and GP100 pAP7 (*bbk32*::Str^R with *bbk32*-R248A/K327A [DA] complement) were incubated with rabbit anti-*B. burgdorferi* antibody coupled with C5-depleted serum. Cells were then probed with murine anti-C4c, followed by anti-mouse Cy5. Cells were fluorescently imaged via confocal microscopy and *B. burgdorferi* antibody binding and C4c deposition were assessed. Cells that have increased colocalization of GFP together with C4c are depicted in light blue in the lower row of each strain tested.

same analysis with the non-borrelial isotype control antibody showed no colocalization of GFP and C4c signal, corroborating the images in S3 Fig.

We then determined if BBK32 protected *B. burgdorferi* from borrelial-specific antibodies and complement-mediated killing in these same infectious isolates and, independently, in an additional parallel set of strains using a quantitative readout. Infectious B31-A3-GFP and ML23 pBBE22*luc* background strains were used for *in vitro* assays in which *B. burgdorferi* were incubated with a targeted polyclonal rabbit antibody against *B. burgdorferi*, coupled with normal human serum (NHS), as a source of complement [42]. Without the addition of the anti-*B. burgdorferi* antibody, both ML23 pBBE22*luc* and B31-A3-GFP are resistant to human serum, as expected based on prior studies determining serum resistance for infectious *B. burgdorferi* (Fig 6) [17]. As a control for the dependence on borrelial-specific antibody recognition, we used the same rabbit isotype antibody for the fluorescent microscopy (Fig 5), together with NHS, and observed no killing of the spirochetes (Fig 6). However, when all strains were incubated with both NHS and the anti-*B. burgdorferi* antibody, only the derivatives lacking *bbk32* were significantly decreased in their ability to survive antibody-dependent, complement-mediated killing (Fig 6). This phenotype was rescued when native *bbk32* was complemented in the *bbk32* mutant strains, but not in isolates expressing the *bbk32*-R248A-K327A allele (Fig 6). The relative difference between the parent, mutant, and complement strains was comparable in both the B31-A3-GFP and ML23 pBBE22*luc* backgrounds (Fig 6), confirming this phenotype is consistent across independently obtained *B. burgdorferi* B31 isolates.

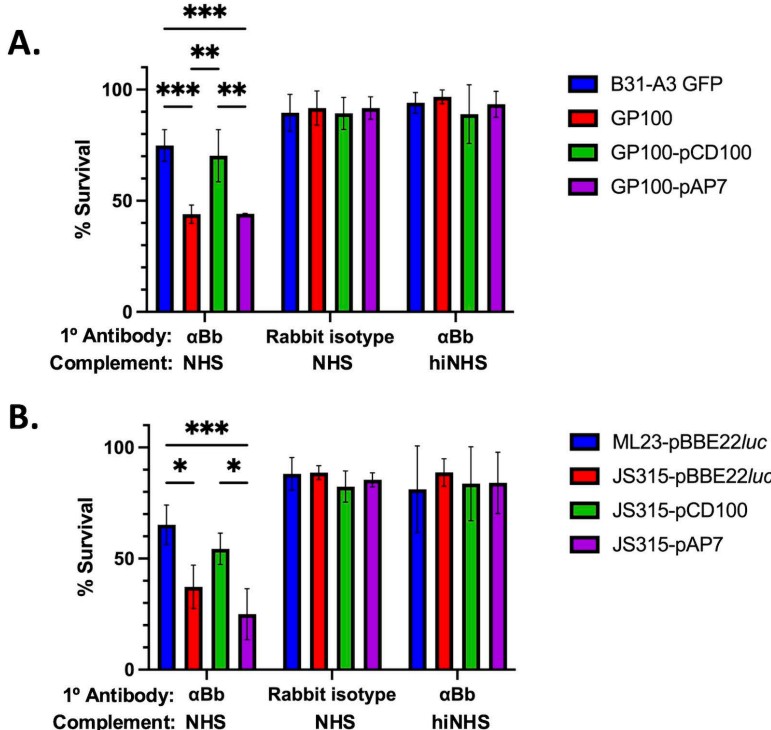

**Fig 6. Antibody-dependent complement-mediated killing of *B. burgdorferi* strain B31-A3 derivatives.** A. The B31-A3-GFP parent strain, the isogenic *bbk32* mutant GP100 (B31-A3-GFP *bbk32*::Str^R), the GP100 strain complemented with either native *bbk32* (GP100 pCD100) or *bbk32*-R248A/K327A double alanine (DA) mutant (GP100 pAP7) were each separately incubated with anti-*B. burgdorferi* antibody coupled with NHS, rabbit isotype control antibody coupled with NHS, or anti-*B. burgdorferi* antibody coupled with heat-inactivated NHS. B. Identical assays were performed for the ML23 derivatives (see Table 1 and Fig 3). Viability of each data set was then assessed via dark-field microscopy based on cell motility and overt membrane disruption in triplicate. Error bars represent standard deviation values. * P < 0.05; ** P < 0.01; *** P < 0.001.

## Discussion

BBK32 is known to contribute to *B. burgdorferi*'s ability to establish experimental infection, thought in part to be due to its ability to bind ECM components and, more recently, its ability to inhibit complement activation [45,46,48–52,55]. The temporal balance of these two functions is unknown, and the multifunctional role of BBK32 during the dissemination phase of infection, where extravasation and complement inhibition are both crucial, has not been explored. Though the impact of normal human serum on a non-infectious isolate of *B. burgdorferi* had been characterized in our previous work [44], the survival of an infectious derivative of *B. burgdorferi* in the presence of a borrelial-specific antibody had not been assessed. We hypothesized that survival of infectious *B. burgdorferi* following exposure to antibody against borrelial antigens—and subsequent activation of classical complement—would be enhanced by the presence of functional BBK32.

As an extracellular pathogen, *B. burgdorferi* is found in blood and interstitial fluid that contain innate immune compounds and cells. Early in infection, and as the infection proceeds, some *B. burgdorferi* are processed by the host, resulting in a potent antibody response against them. Despite a strong humoral response to *B. burgdorferi,* as well as the presence of host complement, *B. burgdorferi* persists, in part, by quelling classical pathway activation despite borrelial antibody-antigen complex formation [24,72]. One additional question that has not been explored is whether BBK32 can bind the C1 complex and ECM components (namely Fn) simultaneously. Our analysis here of infectious isolates with fully functional BBK32, or a BBK32 mutant deficient in classical complement inhibition, is designed to serve as an *in vitro* proxy for testing this possibility.

The work described herein portrays BBK32's C1r-inhibitory activity in a novel context: protection against targeted antibody-dependent complement killing, which occurs concurrently with BBK32's Fn-binding activity. The use of the impaired *bbk32* DA complement afforded us a unique strategy for characterization of a multifunctional protein in *B. burgdorferi.* Namely, normal Fn binding but with an abrogated C1r interaction. We correlated native *bbk32* expression with decreased localization of complement components to the *B. burgdorferi* surface and showed the protective impact of BBK32 against *B. burgdorferi*-specific antibody-directed complement-mediated killing. This analysis provides the foundation for pointed structure-guided mechanistic characterization of other C1r inhibitors—as well as Fn binding—across Lyme disease and tick-borne relapsing fever (TBRF) spirochetes [43,63].

Orthologues to *bbk32* have been identified in TBRF *Borrelia*, some of which have already been determined to bind both Fn and complement components similar to BBK32 [43,63,73]. By leveraging our understanding of the amino- and carboxy-terminal activities of these proteins, we can begin to ask how these multifunctional domains contribute to infection in the face of natural and targeted antibody responses. *B. miyamotoi* FbpA is an example of a C1r inhibitor in a TBRF spirochete that also interfaces with Fn [63]. Through the identification of key "hotspot" residues that mediate BBK32's and *B. miyamotoi* FbpA's interactions with C1r, we have been able to define a consistent, borrelial species-agnostic mechanism for this class of complement inhibitors for a subset of TBRF FbpA proteins [43,63]. Interestingly, independent of their ability to bind Fn, all the C1r inhibitor proteins characterized to date—either BBK32, BBK32-like, or Fbp proteins—contain a disordered N-terminal domain and an ordered C-terminal domain, the latter binding to and mediating the inhibition of mammalian C1r [44–46,63]. Mutations that abrogate C1r binding and inhibition allow for an improved analysis of Fn binding in *B. burgdorferi* pathogenesis could be leveraged for determining the role of the distinct domains found within BBK32. In this study, we have elucidated the ability of BBK32 to bind to both C1r and Fn concurrently. However, a targeted mutagenesis approach is needed to assess how these domains affect borrelial pathogenesis. While we already have mutations in BBK32 that abrogate C1r binding and inhibition, we lack a full-length mutant that reduces or eliminates BBK32-mediated Fn binding. Future studies will initially focus on mutagenizing BBK32's Fn-binding domain [49,50,57,67] to determine if Fn and/or C1r-binding activities promote pathogenic outcomes associated with borrelial infection. Additional recognition of GAGs coincident with Fn or C1r binding seems likely but will require additional experimentation.

There are several pathogens, notably *Staphylococcus aureus*, that produce surface proteins that interact with host ECM and complement components [67,74–76]. This common dual function feature of ECM and complement interaction by additional pathogenic bacteria suggests that this correlation between pathogens is not random and is involved in mediating their pathogenic potential [77,78]. The idea that both distinct functional domains of these proteins might be contributing to immune evasion seems plausible, especially considering Fn's immunomodulatory activity through its ability to bind C1q's collagen-like tail. This Fn::C1q interaction is thought, in some instances, to alter the phagocytosis of pathogens and promote their recognition by phagocytes [79–81]. As this relates to *B. burgdorferi*, BBK32 does not interact with Fn in regions thought to be recognized by phagocytes; thus, it is unlikely that it functions to prevent Fn-mediated phagocytosis by occluding this site. However, BBK32 could be positioning Fn to interact with C1q's collagen tails and sequestering C1q in complex with Fn, limiting the recognition of the C1 complex by host C1q receptors. This type of beneficial immune evasion tactic might also explain BBK32's ability to bind to the C1r zymogen, which previously seemed counterintuitive to the goal of preventing C1r/C1 complex localization to the *B. burgdorferi* surface [44,46,79,80]. Nonetheless, this theory would need to be further investigated.

It is known that infectious *B. burgdorferi* is resistant to human complement due to the presence of numerous genes encoding proteins that inhibit the various complement pathways [26–28,82]. The best characterized of these proteins are the factor H binding proteins that destabilize and inhibit C3 convertase formation and provide resistance to the alternative pathway of complement [31,32,83–86]. Given that BBK32 is an inhibitor of the classical complement cascade, we tested *bbk32* mutants relative to the infectious parent strain for their ability to bind to immobilized C1. We found that there were no differences between these two strains, implying that there were compensatory borrelial proteins that could also bind C1

and potentially inhibit the classical complement pathway [46]. Recently, Pereira et al. used a lipoprotein library to identify novel proteins that are capable of binding to C1 [42] and found, that, in addition to BBK32, the predominantly cp32-encoded Elp proteins recognized C1, specifically C1s [42]. The borrelial genome encodes at least five Elp paralogues that share 44–59% identity and 59–76% similarity. Determining how these proteins augment the classical complement inhibition in borrelial cells—relative to BBK32 function—is not well-characterized, but an area that we are currently evaluating.

The work described in this study was limited to *in vitro* analyses that tracked the role of BBK32 in survival of *B. burgdorferi* via the classical pathway. All the assays used employed borrelial antibodies and active classical complement component to assess damage to borrelial cells using flow cytometry (Fig 4), C4c deposition by fluorescence microscopy (Fig 5) and *in vitro* killing of *B. burgdorferi* infectious strains and derivatives (Fig 6). The data obtained indicates that functional BBK32 is needed for enhanced survival under these conditions. It is important to note that the effects observed in the infectious parent relative to the *bbk32* mutant are not absolute for any of the assays conducted (Figs 5 and 6). This is likely due to compensatory Elp function or other unknown factors that contribute to classical complement resistance.

For most of these approaches the complementation of the *bbk32* mutant with shuttle vectors was achieved with intact *bbk32* or *bbk32*-R248A/K327A sequences (Figs 5 and 6). However, we were not successful using flow cytometry for the *trans* complemented strains perhaps due to enhanced activity associated with the aforementioned Elp proteins. To counteract this limitation, we used a serum sensitive borrelial derivative to produce native BBK32 and separately, a DA mutant variant (i.e., BBK32-R248A/K327A) that was devoid of C1r binding and inhibition, to assess differences in classical complement activation. The results support a role for intact BBK32 to reduce early steps in classical complement activation relative to the DA mutant form (Fig 4). Specifically, the native BBK32 significantly reduced deposition of C4 (in the form of C4c) on the surface of borrelial cells required for forming C3 convertase and promoting downstream proteolytic events that can result in the formation of the membrane attack complex and cell death.

We attempted infectivity studies with the parent, *bbk32* mutant, and the two complemented strains, and the known attenuated phenotype of the *bbk32* mutant was retained [47,52]. However, complementation was not observed, presumably due to inadequate *in vivo* selection of the *trans* complemented *bbk32* alleles or dysregulation of the multicopy *bbk32* locus relative to native *bbk32* expression. Dysregulation of *bbk32* under these conditions might result in borrelial cells that are unable to survive *in vivo* for reasons that are not clear. We are in the process of introducing *bbk32* and, separately, the DA allele in single copy, under control of the *bbk32* native promoter, in the borrelial genome, to retest experimental infection of *B. burgdorferi* with these strains. Another confounding variable for BBK32-related in vivo analysis is the presence of other C1-inhibitors, namely Elp proteins that also inhibit the classical pathway [40–42]. Thus, isolating mutants in multiple genes simultaneously may be necessary to observe a reduced infectivity phenotype, at least in the context of classical complement inhibition. With the growing number of complement-inhibitory genes identified in borrelial species [30,31,40,41,63,87–92], multi-gene knockouts or knockdowns are timely, and should provide clarity about the role of complement inhibition in borrelial survival following infection. The advent of CRISPR-Cas9 systems facilitates approaches to simultaneously knock down or inactivate multiple targets in various combinations [93–96]. These and related studies should define which borrelial complement inhibitory protein(s) are essential for borrelial pathogenic readouts.

In summary, we utilized a *bbk32* mutant and a strain expressing a *bbk32* allele deficient in C1r inhibition in infectious *B. burgdorferi* to assess the spatial dynamics of BBK32's interaction with ECM and complement components. We also demonstrated that BBK32 confers protection from borrelial antibody-initiated complement-mediated damage by flow cytometry, fluorescent microscopy, and whole cell sensitivity assays as readouts. Regarding antibody-dependent, complement-mediated killing, BBK32 could serve to abrogate the deleterious effect of borrelial antigen-antibody interactions that increase as the infection proceeds and class switching occurs. Through our previous structure-guided analyses that identified key residues of BBK32 that interface with C1r [44], we used biophysical analyses to determine BBK32's ability to bind Fn and the C1 complex concurrently *in vitro*, and cell-based experiments to assess these residues' role in antibody-dependent complement protection. This work also describes a strategy that could be applied to

other multifunctional proteins in the borrelial C1r-inhibitor group, namely TBRF pathogens. Taken together, these studies contribute to our understanding of borrelial surface-expressed proteins capable of interacting with distinct host targets and provide a foundation for deciphering the importance of BBK32-mediated Fn binding relative to classical complement resistance in the context of *B. burgdorferi* pathogenesis.

## Supporting information

**S1 Fig. Assessment of AlphaFold3 model quality.** The per residue confidence metric predicted local distance difference test (pLDDT) values are shown on the model using AlphaFold3's standard coloring scheme. pTM: Predicted template modeling score. ipTM: interface predicted template modeling score. The predicted aligned error (PAE) plot is shown on the right with BBK32 corresponding to residues 1–218, Fn to residues 219–397, and C1r residues to 398–637.
(TIF)

**S2 Fig. All *B. burgdorferi* isolates uniformly produce GFP.** *B. burgdorferi* strains B31-A3 GFP, GP100 (B31-A3 GFP *bbk32*::Str$^R$), GP100 pCD100 (*bbk32*::Str$^R$ with native *bbk32* complement), and GP100 pAP7 (*bbk32*::Str$^R$ with *bbk32*-R248A/K327A [DA] complement) were fixed as previously described and imaged via confocal microscopy in the GFP and DIC channels.
(TIF)

**S3 Fig. Isotype control immunoglobulin does not activate the classical pathway in infectious *B. burgdorferi*.** *B. burgdorferi* strains B31-A3 GFP, GP100 (B31-A3 GFP *bbk32*::Str$^R$), GP100 pCD100 (*bbk32*::Str$^R$ with native *bbk32* complement), and GP100 pAP7 (*bbk32*::Str$^R$ with *bbk32*-R248A/K327A [DA] complement) were incubated with an anti-rabbit isotype control antibody coupled with C5-depleted serum. Cells were then probed with murine anti-C4c, followed by anti-mouse Cy5. Cells were fluorescently imaged via confocal microscopy and the degree of rabbit isotype antibody-dependent C4c deposition was assessed.
(TIF)

**S4 Fig. Quantitative assessment of increased deposition of complement components in *bbk32* mutant strains.** Five images of *B. burgdorferi* (one representative image of each group is represented in Fig 5) were scored for the colocalization of GFP and C4c using the integrated coefficient quotient (ICQ) analysis as indicated in the methods. The ICQ for each group is plotted for cells treated with C5-depleted NHS (C5-depl NHS) and either antibody against *B. burgdorferi* or the rabbit isotype control. * $P < 0.05$.
(TIF)

**S5 Fig. Total blot images cropped for Fig 3A.**
(TIF)

**S6 Fig. Total blot images cropped for Fig 3B.**
(TIF)

**S1 Data. Excel spreadsheet with numerical data and statistics for Figs 4, 6A, 6B and S5.** Data for each figure is shown in individual tabs.
(XLSX)

## Acknowledgments

We thank Dr. Patti Rosa and Dr. M. A. Motaleb for providing the infectious *B. burgdorferi* B31-A3 GFP strain. We also are grateful to Haley Przespolewski and Ayesha Nagaria for excellent technical assistance. The authors acknowledge the assistance of Ms. Robbie Moore at the College of Medicine Flow Cytometry and Cell Sorting Facility as well as the

assistance of Dr. Malea Murphy at the Integrated Microscopy and Imaging Laboratory within the Texas A&M College of Medicine (RRID:SCR_021637).

## Author contributions

**Conceptualization:** Brandon L. Garcia, Jon T. Skare.

**Data curation:** Alexandra D. Powell-Pierce, Charles E. Booth Jr., Payton G. Smith, Brittany L. Shapiro, Shannon S. Allen, Brandon L. Garcia.

**Formal analysis:** Alexandra D. Powell-Pierce, Charles E. Booth Jr., Payton G. Smith, Shannon S. Allen, Brandon L. Garcia, Jon T. Skare.

**Funding acquisition:** Brandon L. Garcia, Jon T. Skare.

**Investigation:** Alexandra D. Powell-Pierce, Charles E. Booth Jr., Payton G. Smith, Brittany L. Shapiro, Shannon S. Allen.

**Methodology:** Alexandra D. Powell-Pierce, Charles E. Booth Jr., Brittany L. Shapiro, Shannon S. Allen, Brandon L. Garcia.

**Resources:** Brandon L. Garcia, Jon T. Skare.

**Supervision:** Brandon L. Garcia, Jon T. Skare.

**Validation:** Alexandra D. Powell-Pierce, Charles E. Booth Jr., Payton G. Smith, Brittany L. Shapiro, Shannon S. Allen.

**Writing – original draft:** Jon T. Skare.

**Writing – review & editing:** Alexandra D. Powell-Pierce, Charles E. Booth Jr., Payton G. Smith, Brittany L. Shapiro, Shannon S. Allen, Brandon L. Garcia, Jon T. Skare.

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
