## [Decision Letter · Decision Letter 0]

PPATHOGENS-D-25-01145

BBK32 attenuates antibody-dependent complement-mediated killing of infectious Borreliella burgdorferi isolates

PLOS Pathogens

Jon,

Thank you for submitting your manuscript to PLOS Pathogens. Your manuscript was reviewed by members of the Editorial Board and five external referees. In a testament to this work, it is rare to have so many reviewers agree to the invitation so quickly. Although there is a difference of opinion among the reviewers, most are supportive of your study. Therefore, we invite you to submit a revised version of the manuscript that addresses the points raised during the review process. No additional experimentation is required but based on the reviewers' comments some points need to be presented or explained more thoroughly.

Please submit your revised manuscript within 30 days Aug 03 2025 11:59PM. If you will need more time than this to complete your revisions, please reply to this message or contact the journal office at plospathogens@plos.org. Please include the following items when submitting your revised manuscript:

We look forward to receiving your revised manuscript.

Kind regards,

Jenifer Coburn, PhD

Academic Editor

PLOS Pathogens

D. Scott Samuels

Section Editor

PLOS Pathogens

Sumita Bhaduri-McIntosh

Editor-in-Chief

PLOS Pathogens

orcid.org/0000-0003-2946-9497

Michael Malim

Editor-in-Chief

PLOS Pathogens

orcid.org/0000-0002-7699-2064

**Journal Requirements:**

- TM on page: 12.

4) We have noticed that you have uploaded Supporting Information files, but you have not included a complete list of legends. Please add a full list of legends for your Supporting Information files after the references list.

5) We notice that your supplementary Figures are included in the manuscript file. Please remove them and upload them with the file type 'Supporting Information'. Please ensure that each Supporting Information file has a legend listed in the manuscript after the references list.

6) We note that your Data Availability Statement is currently as follows: "All data are presented in the manuscript and and any supporting files. Any additional inquiries can be directed to the corresponding author.". Please confirm at this time whether or not your submission contains all raw data required to replicate the results of your study. Authors must share the “minimal data set” for their submission. PLOS defines the minimal data set to consist of the data required to replicate all study findings reported in the article, as well as related metadata and methods (https://journals.plos.org/plosone/s/data-availability#loc-minimal-data-set-definition).

7) Please provide a detailed Financial Disclosure statement. This is published with the article. It must therefore be completed in full sentences and contain the exact wording you wish to be published.

1) Please clarify all sources of financial support for your study. List the grants, grant numbers, and organizations that funded your study, including funding received from your institution. Please note that suppliers of material support, including research materials, should be recognized in the Acknowledgements section rather than in the Financial Disclosure

2) State the initials, alongside each funding source, of each author to receive each grant. For example: "This work was supported by the National Institutes of Health (####### to AM; ###### to CJ) and the National Science Foundation (###### to AM)."

3) State what role the funders took in the study. If the funders had no role in your study, please state: "The funders had no role in study design, data collection and analysis, decision to publish, or preparation of the manuscript."

4) If any authors received a salary from any of your funders, please state which authors and which funders..

8) Please ensure that the funders and grant numbers match between the Financial Disclosure field and the Funding Information tab in your submission form. Note that the funders must be provided in the same order in both places as well.

9) Please send a completed 'Competing Interests' statement, including any COIs declared by your co-authors. If you have no competing interests to declare, please state "The authors have declared that no competing interests exist". Otherwise please declare all competing interests beginning with the statement "I have read the journal's policy and the authors of this manuscript have the following competing interests:"

**Reviewers' Comments:**

Reviewer's Responses to Questions

**Part I - Summary**

Reviewer #1: The team of the corresponding authors (Garcia and Skare) have published a series of papers demonstrating the anti-complement (classical pathway) functions of BBK32. In this work, their team (Powell-Pierce et al.) identified the C1r-binding sites, independent on the previously reported Fn-binding sites (activities). Leverage on this information, they further attributed this site (and BBK32-mediated C1r-binding activity) to spirochete’s ability to evade antibody-mediated killing. The work was elegantly done and the paper was well written. I only have few suggestions, specifically on discussing this work and previous work to provide the physiological implications of Lyme borreliae pathogenesis:

Reviewer #2: Skare et al. submitted this manuscript, BBK32 attenuates antibody-dependent complement-mediated killing of infectious Borrelia burgdorferi isolates. This study presents a wealth of new information regarding the interaction of BBK32, FN, and C1r. The authors discovered that BBK32 could bind to ECM, fibronectin, and C1r simultaneously to evade antibody-dependent complement-mediated killing. The authors further applied the AlphaFold3 analysis to predict a ternary complex of BBK32. Other techniques, such as flow cytometry, Fluorescent Microscopy, localization Analysis, and Antibody-Dependent Complement-Mediated Killing Assays, are used to delineate further the evasion mechanisms mediated by the virulence factor BBK32.

Reviewer #3: This is a well-written manuscript addressing the question as to whether BBK32, a fibronectin binding protein of Borrelia burgdorferi, can simultaneously bind fibronectin and also complement C1 without losing the capacity to interfere with complement function. This question is relevant to understanding how a single molecule can serve multiple functions to allow spirochetes to disseminate and evade host immunity in the early pathogenesis of Lyme disease. The study builds on previous structural data to predict fibronectin and complement binding sites on BBK32, and makes use of parent, mutant and reconstituted spirochete mutants to show that BBK32 can protect spirochetes from complement-mediated killing in vitro by reducing C4 deposition on spirochetes. The studies performed are sound, impact more than one Borrelia strain, and the results are clear from the data presented. While it would be helpful to show that this occurs in vivo, the multitude of pathways that Borrelia spirochetes use to evade complement-mediated killing will likely lead to inconclusive results unless these other pathways are also interrupted, which is beyond the scope of this study.

Reviewer #4: This study by Powell-Pierce et al examined whether the two distinct functions previously ascribed to the Borrelia burgdorferi protein BBK32, namely binding to fibronectin and complement C1r could happen without interference from each other. Indeed, previously published identification of the relevant binding residues on the molecule (As131-162 and AS 206-354, respectively, and crystal structures of BBK32 = C1r and BBK32-Fn all had indicated that. This was then here confirmed by using alpha-fold predictions and functionally by plasmon resonance and Western blotting experiments with various mutants. In addition, the authors showed using different mutants of BBK32 that complement deposition was reduced when analyzed by flow cytometry (no primary data shown), image analysis (quantification lacking) and functionally by in vitro killing. Those last assays were done without the testing of Fn binding interference.

Overall, the study confirmed their previous findings that BBK32 supports Bb survival in the presence of complement and previous crystal structure analysis and mutant analysis demonstrating that FN and Complement bind to different parts of the protein.

Reviewer #5: This paper continues the past work, mainly by these investigators, into the functionality of the B. burgdorferi protein BBK32 as protection from complement-mediated killing following infection. In addition to the ability to bind C1r, BBK32 also has a dual property in that it binds ECM components including fibronectin. The authors investigated two hypotheses, that BBK32 can simultaneously bind both C1r and fibronectin, and whether BBK32 provided protection against complement activation in the presence of anti-B. burgdorferi antibodies. The data presented predicts by in vitro AlphaFold3 analysis and confirmed by surface plasmon resonance that dual binding can occur. The authors proceed with experiments designed to demonstrate selective binding of C1r and fibronectin to various wild type and BBK32 mutated proteins and two B. burgdorferi parent strains by Far Westerns followed by assessments of complement activations by flow cytometry and immunofluorescent microscopy. By the results of these experiments, the authors conclude that BBK32 attenuates antibody-mediated complement killing of B. burgdorferi which had not previously been identified.

**Part II – Major Issues: Key Experiments Required for Acceptance**

Reviewer #1: (No Response)

Reviewer #2: The authors have performed SPR using the full-length FN and C1R. Since the authors have reported the BBK32 binding sites of Fn (amino acid residues 131-162) and C1r (206-354), it may be helpful to include the binding domain in the performance of SPR.

The authors may include isothermal titration calorimetry (ITC) further to prove the interaction of Bkk32 with FN and C1r, as it is a valuable technique for studying protein-protein interactions using both surface plasmon resonance (SPR) and ITC.

Reviewer #3: The authors have provided a description of additional studies that are either in progress or would be necessary to have a more complete story of how BBk32's dual role as an adhesin and as an inhibitor of complement contributes to Borrelia burgdorferi pathogenesis. However, I believe this manuscript is a complete enough story as is and no additional experiments are necessary for this to be published.

Reviewer #4: 1) The data in Figure 4 cannot be verified without showing the flow cytometry results.

2) The imaging results for C4c deposition on Bb requires proper image quantification to normalize for the presence of Bb. It is common to also provide a darkfield image to confirm the presence of gfp expression on all spirochetes. Indication should be provided about the number of spirochetes analyzed and the frequency of those showing deposition. If deposition is reduced but not eliminated, then an average MFI might be used instead.

3) It is unclear why figures 4-6 were done exclusively testing of complement inhibition, which had been previously amply demonstrated, when the novel information to be provided in this manuscript was to explore the interaction of Fn-C binding simultaneously.

Reviewer #5: No major issues

**Part III – Minor Issues: Editorial and Data Presentation Modifications**

Reviewer #1: 1. Before the authors’ team identified BBK32 as a C1r-binding protein (Garcia et al. PLoS Pathog 2016), BBK32 was known to have two functions (glycosaminoglycan (GAG)-binding activity and fibronectin (Fn)-binding activity). Lin et al. Cell Microbiol 2015 from Dr. John Leong’s team and Moriarty et al. Mol. Microbiol. 2012 from Dr. George Chaconas’s lab located where those activities are on the BBK32 (residues 45-68 and 158 to 182 (and 182 to 209) of BBK32). Lin et al. and Moriarty et al. have used the spirochetes producing BBK32 with internally deletions on the amino acids 45-68 and 158-182 to attribute Fn- and GAG-functions of BBK32 to vascular interactions (through short term i.v. injection of mice) and early stages joint colonization (through low dose, intradermal inoculation of mice). Therefore, the information described in the line 115-116 and 433-434 may not be accurate, especially to reflect the findings of those papers.

2. The authors have identified BBK32 to be able to simultaneously bind to Fn and C1r, suggesting that the functions of Fn-binding and C1r-binding of BBK32 can be performing simultaneously during infection. GAG binds to amino acids 45-68 of BBK32, far from Fn and C1r-binding sites. Therefore, although the authors did not characterize whether BBK32 can bind to Fn, C1r, simultaneously with GAG, it is very likely that BBK32 can bind to these three ligands simultaneously. As Lin et al. 2015 and Moriarty et al. 2012 have defined the functions of GAG and Fn for BBK32 during infection, the physiological implications in the case that BBK32 evolved to bind to three ligands at the same time would be worth to discuss. Lin et al. and Moriarity et al. each proposed a model to explain the functions of GAG and Fn at early stages of infection. C1r seems to be the last piece of puzzle of the picture to elucidate the role of BBK32 during early infection.

3. The papers from one of the corresponding authors in this manuscript, Dr. Skare, (Seshu et al. Mol. Microbiol. 2006 and Hyde et al. Mol. Microbiol. 2011) showed that a bbk32 mutant (the strain JS315 in ML23 missing lp25) when introduced into the mice under low dose close to ID50 (10^3 per mouse) displayed partial colonization defects at early stages of infection. This work later was later confirmed by this manuscript and Lin et al. Cell Microbiol. 2015, indicating the strength of controlling the dose under intradermal injection to delineate the roles of functionally redundant Lyme borreliae proteins. However, Li et al. from Dr. Erol Fikrig’s laboratory showed that a bbk32 mutant in the background of a bacterial strain with lp25 does not have any defects in mouse-to-tick acquisition, tick-to-mouse transmission, and tissue colonization (Li et al. Infect Immun 2006). It would be worth to discuss these results of BBK32 from the perspectives of Lyme borreliae pathogenesis. Such discussions would be helpful because it has been very common for the strain deficient of different Lyme borreliae gene (e.g., DbpA, CspZ, CspA) showing different phenotypes under tick vs. needle infection models.

4. The authors mentioned that some partial phenotypes in this study could be due to the compensatory functions of Elp (line 476-477). Does the background strain that is in this study encode Elp?

Reviewer #2: Fig. 3: It would be helpful to have a band density plot using ImageJ. The present legend is not easy to understand. It may be useful to provide a brief explanation in the legend.

Fig. 5: It may be helpful to include an arrow in the overlay and provide a brief explanation in the legend.

Reviewer #3: This manuscript is clearly written. The only modification suggested is in the Discussion section, lines 394-397, regarding the stated hypothesis that survival of B. burgdorferi after exposure to Borrelia antibodies and complement would be diminished in the presence of functional BBK32. If this is indeed the hypothesis, then there should be another sentence indicating that their results are opposite to this due to the ability of BBK32 to bind both fibronectin and inhibit complement activation simultaneously.

Reviewer #4: 1) Studies by John Weis (Jacobson et al. 2007) and others showed that complement receptor-mediated killing is not required for control of Borrelia by mice. Thus, the extent to which complement can opsonize to support Borrelia-killing is unclear. A point that should be considered and acknowledged.

Reviewer #5: 1. As a personal opinion, I prefer the Supplemental Figures 3 and 4 over Figure 3. Showing the entire blots rather than selective cutaways has always seemed more believable (at least to me). Plus, the blots are beautifully clean from non-specific binding. Just a note for consideration.

Another small note; the authors could have considered adding a recombinant BBK32 lane to the blots to ensure that the fibronectin and C1r were indeed binding to native BBK32 in the lysate lanes instead of the possibility of binding to a different protein that co-migrates with BBK32 in the gel.

2. Regarding the fluorescent microscopy images in Fig. 5, hopefully it would be possible to lighten the brightness on the images especially for the anti-C4c pictures. Even on the computer screen, it was difficult to compare those images with the overlay images. So the statement in the text Results lines 357-362 that more C4c was seen in the mutant strains than the parents doesn’t look convincing and is debatable. Perhaps lightening the images and adding arrows to show borrelia without C4 staining in the parents could be helpful. The flow data from Fig. 4 is more convincing.

3. Strictly an editorial comment. The authors provide a comprehensive background narrative in the Introduction and the Discussion. However, although nicely structured and informative, the entire text (mainly in the Results) could be improved for reading by eliminating needless words and phrases. There are several instances of: “to (or towards) this end…”, “in order …”, “…we then…” , “to begin…”, “we sought to…”; “ we next…” that when removed streamlines the reading.

Line 69; “these hosts” may be better than “organisms”

Line 102; “pathways” (plural)

Lines 107-109; sentence repeats “classical pathway”. Could be rewritten as “ BBK32, which is upregulated during vertebrate infection, can potently block activation of the classical pathway of complement by binding to C1r ”.

Lines 277-279; this sentence redundant to line 272 that starts this paragraph. Can be omitted.

PLOS authors have the option to publish the peer review history of their article (what does this mean? ). If published, this will include your full peer review and any attached files.

**Do you want your identity to be public for this peer review?** For information about this choice, including consent withdrawal, please see our Privacy Policy .

Reviewer #1: No

Reviewer #2: No

Reviewer #3: No

Reviewer #4: No

Reviewer #5: No

**Figure resubmission:**
---

## [Editor Report · Decision Letter 1]

Jon,

Thank you for submitting your revised manuscript with detailed point-by-point responses to the Reviewers' comments. We are pleased to inform you that your manuscript 'BBK32 attenuates antibody-dependent complement-mediated killing of infectious Borreliella burgdorferi isolates' has been provisionally accepted for publication in PLOS Pathogens.

Best regards,

Jenifer Coburn, PhD

Academic Editor

PLOS Pathogens

D. Scott Samuels

Section Editor

PLOS Pathogens

Sumita Bhaduri-McIntosh

Editor-in-Chief

PLOS Pathogens

orcid.org/0000-0003-2946-9497

Michael Malim

Editor-in-Chief

PLOS Pathogens

orcid.org/0000-0002-7699-2064
---

## [Editor Report · Acceptance letter]

Dear Dr. Skare,

We are delighted to inform you that your manuscript, "BBK32 attenuates antibody-dependent complement-mediated killing of infectious Borreliella burgdorferi isolates," has been formally accepted for publication in PLOS Pathogens.

Best regards,

Sumita Bhaduri-McIntosh

Editor-in-Chief

PLOS Pathogens

orcid.org/0000-0003-2946-9497

Michael Malim

Editor-in-Chief

PLOS Pathogens

orcid.org/0000-0002-7699-2064